# Transcriptomic Insights on the Preventive Action of Apple (cv Granny Smith) Skin Wounding on Superficial Scald Development

**DOI:** 10.3390/ijms222413425

**Published:** 2021-12-14

**Authors:** Nadia Cainelli, Cristian Forestan, Dario Angeli, Tomas Roman Villegas, Fabrizio Costa, Alessandro Botton, Angela Rasori, Claudio Bonghi, Benedetto Ruperti

**Affiliations:** 1Dipartimento di Agronomia, Animali, Alimenti, Risorse Naturali e Ambiente, Università di Padova, 35122 Legnaro, PD, Italy; nadia.cainelli@phd.unipd.it (N.C.); alessandro.botton@unipd.it (A.B.); angela.rasori@unipd.it (A.R.); claudio.bonghi@unipd.it (C.B.); 2Dipartimento di Scienze e Tecnologie Agro-Alimentari, Università di Bologna, 40127 Bologna, BO, Italy; cristian.forestan@unibo.it; 3Fondazione Edmund Mach, Centro di Trasferimento Tecnologico, 38010 San Michele all’Adige, TN, Italy; dario.angeli@fmach.it (D.A.); tomas.roman@fmach.it (T.R.V.); 4Centro Agricoltura Alimenti Ambiente, 38098 San Michele all’Adige, TN, Italy; fabrizio.costa@unitn.it

**Keywords:** apple superficial scald, post-harvest, cold storage, abiotic stressjasmonic acid, ethylene, senescence, RNA-Seq, transcriptomics

## Abstract

Superficial scald is a post-harvest chilling storage injury leading to browning of the surface of the susceptible cv Granny Smith apples. Wounding of skins has been reported to play a preventive role on scald development however its underlying molecular factors are unknown. We have artificially wounded the epidermal and sub-epidermal layers of apple skins consistently obtaining the prevention of superficial scald in the surroundings of the wounds during two independent vintages. Time course RNA-Seq analyses of the transcriptional changes in wounded versus unwounded skins revealed that two transcriptional waves occurred. An early wave included genes up-regulated by wounding already after 6 h, highlighting a specific transcriptional rearrangement of genes connected to the biosynthesis and signalling of JA, ethylene and ABA. A later transcriptional wave, occurring after three months of cold storage, included genes up-regulated exclusively in unwounded skins and was prevented from its occurrence in wounded skins. A significant portion of these genes was related to decay of tissues and to the senescence hormones ABA, JA and ethylene. Such changes suggest a wound-inducible reversed hormonal balance during post-harvest storage which may explain the local inhibition of scald in wounded tissues, an aspect that will need further studies for its mechanistic explanation.

## 1. Introduction

Apples are subjected to prolonged periods of storage at low temperatures to delay ripening and senescence and prolong their marketing period. Even though apple fruits can withstand relatively long exposures to low nonfreezing temperatures, extended periods of cold storage are often linked to the development of physiological disorders. Indeed, prolonged cold exposure represents a stress situation for apples and often leads to chilling injuries that may occur in a mostly unpredictable way, depending on several factors such as the intensity and duration of cold exposure, the specific genotypic sensitivity of the fruits and their developmental ripening stage at harvest in combination with several pre-harvest factors [1]. One of the most important and most studied chilling injuries of apples is superficial scald due to its wide economic impacts in susceptible varieties (e.g., Granny Smith). Superficial scald is finally manifested with the necrosis within portions of the fruit epicarp and the first 5–6 hypodermal cell layers leading to the appearance of slightly sunken irregularly distributed brown skin patches on the surface of the fruits [2,3]. Superficial scald symptoms become manifest mainly during shelf life at room temperature (20 °C) following a period of cold exposure (−1 to 4 °C) of at least two-three months [2,4] and can be commercially controlled by different chemical treatments such as the application of antioxidants such as diphenylamine (DPA), 6-ethoxy-1,2-dihydro-2,2,4-trimethyl quinoline (Ethoxyquin), by treatment with the ethylene inhibitor 1-methylcyclopropene (1-MCP, 0.5 μL L^−1^) and/or by storage in low oxygen (LO_2_, <0.5%) atmosphere [5]. For these reasons scald is considered an oxidative phenomenon that largely depend on ethylene action. Despite its economic, agronomic and scientific importance, knowledge regarding the underlying molecular processes that modulate the onset of the scald phenomenon during the cold stress exposure of fruits are still not fully clarified.

Several studies have shown that apple scald is overall an oxidative phenomenon and, in fact, related to the production of conjugated trienol products, arising from the oxidation of the sesquiterpene *α*-farnesene, which accumulate during storage in response to cold stress. The synthesis of *α*-farnesene is closely linked to ethylene biosynthesis and action [6], since this hormone modulates the expression of the gene MdAFS1 encoding *α*-farnesene synthase 1, the last enzyme in the *α*-farnesene biosynthetic pathway [7].

From the molecular point of view, recent transcriptomic and metabolomic analyses showed an important change in polyphenolic profiles, characterized by the accumulation of chlorogenic acid, taking place during the shelf-life phase and during symptoms development [8]. Chlorogenic acid can be further oxidised by polyphenol oxidase (MdPPO) which gene, according to several authors, is highly up-regulated during cold stress exposure and subsequent shelf life [2,9,10]. Recent studies have exploited omic approaches to uncover the overall molecular and metabolic changes that occur during both storage and shelf-life and are associated or not with scald development [3]. Most of these studies, aimed at the systematic identification of molecular and metabolic changes taking place during the triggering phase of scald induction and its subsequent phase of symptoms development, have relied on the use of DPA (or antioxidants) and 1-MCP (ethylene inhibition) as tools to create experimentally separate groups of apples characterised by a low potential for scald development as opposed to groups of apples with a high potential for scald development. In a recent study, ozone (O_3_) exposure was identified as a scald-inducer and was exploited as an additional approach to study the molecular control of scald. However also in this latter case ethylene was identified as a key regulatory factor mediating the scald inductive action of O_3_ [10]. The identification of experimental systems that may enable to artificially control scald prevention or induction, possibly independently of ethylene signalling, will be of great importance for the identification of the molecular mechanisms underlying its induction and symptoms development.

In a seminal article from Abdallah et al. (1997) it was reported, based on previous experimental observations, that wounding or mechanical damage inhibited apple superficial scald formation in the close vicinity of the wounded sites. The authors suggested that the lower scald incidence in the wounded areas of apples could be associated with an increase in conjugated trienes and total glycolipid content, together with a decreased proportion of linolenic acid and of the glycolipid unsaturation index of the glycolipid fraction. An increase in cinnamic acid derivatives in the immediate proximity of the wounded area suggested that an upraise of antioxidant activity had been evoked by wounding [11]. This study suggests that wounding, which is generally associated with an oxidative stress and a release of reactive oxygen species (ROS) through the activity of respiratory burst NADPH oxidase homologs (RBOH) [12], may evoke a cascade of signaling events in the close proximity of the wound that result in prevention of scald. In a recent analysis in Granny Smith apples, the ROP-GAP rheostat, which is a key module for the regulation of reactive oxygen species homeostasis through the action of respiratory burst NADPH oxidase homologs (RBOH), has been identified to be up-regulated by 1-MCP as an early event in scald prevention [9]. This evidence taken together may suggest that during the inductive phase of scald ROS levels and their homeostatic maintenance may play a preventive role.

This study was aimed to confirm and further investigate the wound-mediated inhibition of scald induction by unravelling the local molecular events triggered in response to wounding and by identifying the putative connections between scald incidence and wounding responses. RNA-Seq analyses were carried out in time-course experiments to systematically identify the genes that were specifically expressed in response to mechanical damage and to identify the wound-induced pathways that may be associated with the wound-dependent prevention of scald.

## 2. Results

### 2.1. Apple Superficial Scald Incidence in Relation to Wounding

In order to evaluate the effects of wounding on superficial scald development, the shelf life of Granny Smith apples was evaluated over two subsequent years (2019 and 2020) at three time points of cold storage following wounding: six hours after wounding (time point 0), after one month and after three months of cold storage in CA (1.7 °C, O_2_ 5%, CO_2_ 1.3%, relative humidity 60/70%). The choice of the time point at six hours was made to highlight the early regulatory transcriptional responses to wounding. The time points after one and three months of storage were chosen to highlight the medium- and long-term transcriptional regulations taking place before and after the induction of superficial scald, respectively. This choice was based on previously published works showing that the cold stress threshold for superficial scald induction is commonly reached by Granny Smith apples after a minimum of two to three months of cold storage [2,10]. At the mentioned time points apples were shifted from 1.7 °C to room temperature and scald development was evaluated over a period of 8 days taking pictures of the whole apple surface after 6 h, 3 days and 8 days of shelf life. Subsequently scald incidence was estimated as a percentage of the total apple surface area by calculating the percentage of brown areas over green areas using the ImageJ online tool. In both years, as expected, no visible scald symptoms were detectable after 8 days of shelf life up to one month of storage while superficial scald symptoms became evident during shelf life after 3 months of cold storage (Figure 1A). During the shelf life following 3 months of cold storage superficial scald was visible on almost all analysed apples (95%) with a mean percentage area interested by scald of 25–30% on control apples and of 15% on wounded apples after 3 days of shelf life, and of 30–50% on control and 20–30% on the wounded half after 8 days of shelf-life, respectively. The means of percentage of scald incidence of the wounded apple half versus the non-wounded half were compared using a t-test. Results showed that after 3, 5 and 8 days of shelf-life scald incidence was consistently significantly lower on the wounded half of the apple. Statistical analysis displayed that the mean scalded area of the wounded half and that of the control half were significantly different at all time-points of shelf life with a *p*-value of 0.05 in 2020 and of 0.1 in 2019, confirming the effect of the vintage on superficial scald incidence, as described by several authors [2] and showing that wounding in fact had an effect of containment on the insurgence and/or spreading of superficial scald (Figure 1A,B), thus confirming the experiments originally described by Abdallah et al. [11].

### 2.2. Comparison of Global Transcriptomic Profiles of Wounded versus Non-Wounded Apple Skins

RNA-Seq analyses were employed to characterise and compare the global transcriptomic profiles of the wounded and non-wounded apple skin tissues during CA post-harvest cold storage in a time-course experiment starting six hours after wounding (6 h), after one month (1 M) (before the visible induction of apple scald development) and three months of storage (3 M) (when the visible induction of apple scald development became apparent). Three biological replicates for each time-point and experimental thesis were used for transcriptomic analysis. After sequencing on a Illumina platform, high-quality reads were mapped on the Malus x domestica GDDH13 v1.1 reference genome [13] (sequencing and mapping statistics are reported in Appendix A). Principal component analysis (PCA) of sequenced samples and replicates was performed on gene read counts to evaluate the biological replicates quality. Differentially expressed genes (DEGs) in wounded versus non-wounded apple skin tissues at each time-point were therefore identified using DESeq2 [14]. At 6 h after wounding, and 1 M and 3 M of storage, 384, 109 and 402 DEGs (log2 fold change >|0.58| -corresponding to a 1.5-fold change variation in expression level—and adjusted *p*-value ≤ 0.05) were identified (Figure 2). Time-point specific DEGs are reported in Appendix A, while normalized mean gene expression values for each gene in the six samples, expressed as Fragments Per Kilobase per Million mapped reads (FPKM) are reported in Appendix A.

For each comparison, the genes showing a log2 fold change ratio >|0.58| (corresponding to a 1.5-fold change variation in expression level) and a false discovery rate (FDR)-adjusted *p*-value ≤ 0.05 were considered as differentially expressed genes (DEGs).

Pairwise comparisons between wounded and control skin tissue at each time-point, considered separately, resulted in the identification of 384 significant DEGs after 6 h, and in 109 and 402 DEGs after 1 and 3 months of storage, respectively (Figure 2). Differential expression of the identified DEGs was validated by real-time RT-qPCR on ten selected differentially expressed genes, showing a 90% confirmation rate of the RNA-Seq data (Appendix A).

After 3 months of storage an important shift in transcriptional regulation was apparent: in fact, 6 h after wounding and 1 M of storage a wide majority (87.8% and 92.7%, respectively) of the DEGs were up-regulated in wounded tissue with respect to the unwounded control, while after 3 M only 9.2% of all significant DEGs were up-regulated resulting in 90.8% of down-regulated genes in response to wounding in comparison to control conditions (Figure 2A). The genes whose transcription was up- and down-regulated were classified and grouped based on the magnitude of their transcriptional changes (Figure 2B). DEGs with a log2FC bigger than 2 were classified as UpClass3 genes (orange), those with a log2FC between 1 and 2 as UpClass2 (yellow), log2FC between 0.58 and 1 as UpClass1 (pale yellow) and genes with a log2FC between 0 and 0.58 were considered as NoVariation class (grey). The same type of classification was adopted for down regulated genes (DownClass 1 (light blue): log2FC between −0.58 and −1; DownClass2 (blue): log2FC between −1 and −2; DownClass3 (dark blue): log2FC smaller than −2). Adopting this classification, after 6 h up-regulated genes were equally divided into the 3 classes (25% UpClass1, 26% UpClass2 and 26% UpClass3) while down regulated genes consisted in a smaller portion (1% DownClass3, 3% DownClass2 and 7% DownClass1). At 6 h 12% of all DEGs showed no variation in expression levels. After 1 month the picture appeared similar: 35% of DEGs were classified as UpClass3, 18% as UpClass2 and 32% as UpClass1, 8% of DEGs showed no variation while DownClass1 and DownClass2 contributed to 3% of DEGs. DownClass3 comprehended only 1% of DEGs. After three months of storage this scenario was mainly reversed, in fact 52% of DEGs belonged to DownClass3 being strongly downregulated, 30% to DownClass2 and 8% to DownClass1 while no DEGs could be attributed to UpClass3 (2% to UpClass2 and 7% to UpClass3). Only 1% of DEGs showed no variation at this time point.

Considering down-regulated DEGs (393) (Figure 2C, right panel) and up-regulated DEGs (459) (Figure 2C, left panel) separately, no down-DEGs were shared between all the considered time-points while only very few up-regulated DEGs were shared (0.4% in the comparison between 6 h and 3 M; 15% between 6 h and 1 M; and only 0.2% between all groups).

Overall, the dynamics of transcriptional changes pointed out that early responses to wounding were associated with a prevalence of a positive transcriptional regulation on a set of specific genes, testified by the large number of up-regulated genes after 6 h, that appeared to be transient since most of these DEGs were uniquely regulated at this time-point. In fact, only three DEGs that were up-regulated after 6 h from wounding remained up-regulated also after one and three months of storage (Figure 2C, left panel). At three months of storage the number of down-regulated genes increased remarkably, suggesting the presence of a triggering point during storage, temporally located between 1 and 3 months, leading to a substantial change in response to wounding that may be associated with the prevention of superficial scald. These data suggest that at least two different waves of transcriptional regulation took place in response to wounding, an early one, most likely linked directly to the effects of mechanical damage (6 h after wounding) and a late one taking place after three months of storage and therefore possibly a consequence of the effects of wounding on overall physiology of the apple skin. This phase change seemed to be associated with a transient de-regulation of gene expression evidenced after one month of storage, supported by the lower number of DEGs at this time point. These data fit also with the temporal dynamics of scald induction reported independently by different authors [2,10].

A GO enrichment analysis was performed on the DEGs identified at the different time-points, to functionally characterize these transcriptional regulatory profiles. At 6 h after wounding (Figure 3, panel A) several biological pathways were found to be enriched in the up-regulated DEGs in the wounded apple half. Concerning the GO category “Biological process”, as expected, the “response to wounding” pathway resulted to be enriched in wounded tissues. Consistently, biological processes that are evidently linked to abiotic/biotic stresses ranked in the first positions: jasmonic acid biosynthetic processes, salicylic acid catabolic processes, response to chitin, defense response to oomycetes and response to stress hormones such as response to abscisic acid and jasmonic acid. Additionally, oxidation-reduction process pathways were found to be enriched among other biological pathways involved in cellular response to hypoxia suggesting also a crosstalk with low oxygen stress. Among down regulated genes in the wounded apple half at 6 h only “proline metabolic process” was found enriched as a biological process.

After 1 month of storage (Figure 3, panel B) GO analysis among up-regulated DEGs pointed out that oxidation-reduction processes, response to wounding and salicylic acid catabolic processes pathways were still enriched in wounded apples. Due to the low number of DEGs after 1 month of storage GO analysis did not produce any result for down regulated genes.

Looking at the enriched biological pathways that were down regulated in wounded tissue after three months (Figure 3, panel C) of conservation oxidation reduction processes, response to wounding, salycilic acid catabolic processes and other different defence responses (response to biotic stimulus, response to bacteria, response to virues and oomycetes) were enriched. This finding suggested that some of the biological processes up regulated after 6 h of wounding were down regulated after three months. Among up regulated DEGs, that were less represented at this time point (3 M), only very few biological pathways such as photosynthesis, protein chromophore linkage and response to light stimuli could be identified as enriched.

### 2.3. Conserved Transcriptional Regulation in Wounded and Unwounded Apple Skins during Storage

Considering the above-described evidence on the occurrence of different transcriptional waves during storage, we aimed at further investigating the transcriptional regulation evoked by wounding over time and its specific effects on the progression of the ripening events normally taking place during storage. For this purpose, time course gene expression analyses were performed on both wounded and not-wounded samples using the likelihood ratio test (LRT) in DESeq2. LRT allows the identification of any gene that shows a change in its expression across the different levels of a factor (timepoints). Generally, this test results in a larger number of genes than the individual pair-wise comparisons between consecutive time-points: consequently only genes showing an adjusted p-value below 0.01 and showing a fold change in expression of at least 2 (up or down) in the comparison of each stage with the remaining two were considered as significantly differentially genes expressed during storage. Genes that were differentially regulated among the different time-points in the wounded or in the control halves of the apples were identified and compared: 7416 genes were identified as DEGs for the wounded half and 7141 DEGs were found for the control skins, respectively, resulting in an almost 1:1 ratio of DEGs.

Hierarchical clustering of these DEGs allowed the identification of four main different expression patterns in both experimental theses (Figure 4, panel A and B) and the following three transcriptional trends appeared conserved between wound and control: (1) genes showing a high expression at the beginning of the experiment (6 h) and a progressive downregulation during storage (1 M and 3 M)(group 1 of control apples and group 2 of wounded apples); (2) genes characterized by a low expression at the beginning of the experiment (6 h) and progressively up-regulated over time (1 M and 3 M)(group 2 control and group 1 wounded); (3) genes displaying a low expression level at 6 h, reaching a transient peak of expression at 1 M followed by a decline after three months of conservation (3 M). Two additional groups of genes were identified that did not share their temporal expression patterns between wounded and control apples: group 4 of wounded apples included genes with an intermediate expression level at 6 h followed by a maximum expression at 1 M and a strong down-regulation at 3 M (Figure 4, panel B); conversely, control apples displayed a group of DEGs with a unique expression profile (group 3) (Figure 4, panel A) that could not be identified in DEGs of wounded apples, characterised by a transient down-regulation after one month and a reactivation of transcription after three months of storage. When the DEGs belonging to groups with similar expression patterns were compared, a partial overlap of DEGs was found as shown in Venn diagrams in panel C (Figure 4) pinpointing conserved and divergent molecular responses between wounded and unwounded areas of the apple skins. Considering the genes progressively up-regulated over time in both control and wounded apple skins, about 50% of the DEGs resulted to be in common while 19.4% were exclusively regulated in wounded tissues and 30% were found to be regulated only in the control skins. In the second comparison (group 2 control vs. group 1 wound) only 32.2% of DEGs were shared between the two groups and 39.6% of DEGs were regulated exclusively in wounded tissue; while for the third comparison 45.1% of DEGs were shared between the two groups and 31.3% of DEGs were included exclusively in wounded apple peel.

For the DEGs that were shared in both control and wounded expression groups (Figure 4, panel C) a GO analysis was performed (Figure 4, panel D) to identify the molecular processes that were conserved and thus, not affected by wounding. The most frequent annotations in the molecular function category in the first comparison (genes progressively up-regulated during storage) were several genes encoding for transporters, reductases and enzymes with phosphatase activities, as well as inositol heptakisphosphate kinase activity and DNA-binding transcription factor activity. For the second comparison (group 2 control and group 1 wound) hydrolase, carboxylase and beta-glucosidase activities were found enriched along with other molecular functions such as sedoheptulose-bisphosphatase activity, unfolded protein binding, protein domain specific binding, cellulose binding and feruloyl esterase activity. In the third comparison (group 4 control and group 3 wound, including DEGs up-regulated and peaking after one month of storage) GO analysis evidenced molecular functions of ethylene receptor activity, ethylene binding and pyruvate decarboxylase activity to be enriched along with phenylalanine ammonia-lyase activity, oxidoreductase activity, monogalactosyldiacylglycerol desaturase activity, galactinol-sucrose galactosyltransferase activity, sulfotransferase activity and phosphorelay sensor kinase activity. Overall, the wide range of metabolic processes that appeared to be shared by both wounded and unwounded skins confirmed that several biochemical pathways were not influenced by wounding.

In the third comparison (Figure 4D, right panel, genes shared by group 3 and 4), noteworthy is the enrichment of genes involved in the ethylene sensing and signal transduction in both wounded and unwounded fruits, suggesting that wounding did not interfere with at least a subset of ethylene signalling events, and thus with at least one part of the ripening process. In particular, the DEGs underlying the GO enriched molecular functions “ethylene binding, ethylene receptor activity” included two genes encoding ethylene receptors (MD03G1292200 and MD06G1001100), indicating the existence of an ethylene signalling module that remained unaffected by wounding and of conserved ethylene responses in both wounded and unwounded portions of the fruit during storage.

### 2.4. Evaluation of the Combined Effect of Wounding and Time of Storage on Transcriptional Dynamics

The previously reported results pointed out that a significant number of genes is similarly regulated in both wounded and control conditions as a result of a partially conserved fruit maturation process during cold storage but, at the same time, also suggested that the time-course of expression of several genes may be specifically influenced by wounding, through wounding-specific signalling events activated on apple skin tissues. The analysis of the molecular responses evoked by wounding over the period of postharvest ripening led to the identification of 448 DEGs that were specifically regulated by wounding over time in comparison to the undisturbed control (Figure 5). By plotting these DEGs on a heat map, a macroscopic difference between the expression patterns appeared particularly evident after 3 months of storage when several DEGs up-regulated exclusively in the control (unwounded) halves of fruits appeared not to be regulated or to be down-regulated in the wounded ones. Similarly, a transient transcriptional up-regulation of DEGs could be evidenced shortly after wounding (6 h) which was absent in control tissues (Figure 5). After 1 month of storage only slight differences in gene expression patterns could be observed. These time-course analyses of the dynamics of transcriptional waves taking place over time confirmed, through an independent approach, the evidence obtained by the static comparison at each time-point conducted previously (Figure 2 and Figure 3).

Analysing the expression dynamics of these 488 DEGs, 6 differential expression clusters could be identified (Figure 6), which partially recalled the previously described patterns of expression of DEGs in the time-course analysis (Figure 4). Wounding specifically affected the expression of these DEGs during storage: for 113 genes (group 1 and 2, Figure 6, panel A) the stable expression pattern over the first two time points (6 h and 1 M) followed by a significant upregulation of gene expression after 3 months (3 M) of storage in control conditions, appeared to be reversed in wounded apples skins, in which the transcriptional level remained either unchanged (Figure 6A, group 2) or subjected to down regulation (Figure 6A, group 1). Similarly, the 289 differentially expressed genes belonging to group 3, 4 and 7, displayed in unwounded skins a transient downregulation at 1 month of storage followed by an upregulation after 3 months of storage (Figure 6, panel B). Conversely, in wounded skin tissues the same DEGs showed a progressive down-regulation over time (Figure 6B, group 4 and 7) or a stable basal expression (Figure 6B, group 3). A further group of genes (Figure 6, panel C, group 6) included DEGs that were progressively downregulated in control conditions and were even further downregulated in wounded tissues. Overall, these data show that wounding evoked a specific signalling cascade that was reflected by substantial changes in the dynamics of expression of specific groups of genes, that is in most cases a transient upregulation after six hours from wounding followed by a more marked downregulation in comparison to control conditions.

GO analysis performed on these expression groups enabled the identification of molecular functional categories enriched as a result of wound-specific effects over storage (Figure 7). Among DEGs for which the up-regulation after 3 months of storage was inhibited by wounding, oxidoreductase and dioxygenase activity, and RNA polymerase activity related to heat stress and cellular responses to heat stress, resulted to be enriched, as well as salycilic acid catabolic process and defence to oomycets (Figure 6, panel A, group 1 and 2; Figure 7, left panel). For the genes included in groups 3, 4 and 7 (Figure 6, panel B)(transiently downregulated at 1 month and upregulated at three months in control apple skins but not in wounded tissues) molecular functions such as response to chitin, to biotic stimulus and to wounding and defence responses, were enriched along with protein kinase activity, protein serine/threonine kinase activity, abscisic acid binding, protein phosphatase inhibitor activity, methyl indole-3-acetate esterase activity and signaling receptor activity (Figure 7, centre panel). It is worth of note that these GO categories consistently and largely overlapped with those that were previously found to be enriched in up-regulated DEGs after 6 h and down-regulated DEGs after three months from wounding (Figure 3, Panel A and C).

### 2.5. Pathway Analysis of Hormonal and Regulatory Genes in Apple Skin Wound Responses

An in depth analysis of the identity of the most differentially expressed genes (log2FC > 3.5 and log2FC < −3.5 respectively) (Table 1) and of DEGs located in specific pathways by MapMan pathway analysis (Appendix A) pointed out that wounding exerted a differential effect on different members of the ethylene biosynthetic and signalling pathway at the different time-points: two genes encoding ethylene response factors ERFs (MD06g1051800; MD11g1210700) and three encoding ERF/AP2 domain transcription factors (MD15g1334900; MD03g1049900; MD05g1030800) and one encoding ACC Oxidase (ACO3, MD17g1106300) were specifically up-regulated after 6h of wounding (Appendix A), one ERF encoding gene (ERF1, MD16G1216900) was exclusively up-regulated in wounded skins after one month of storage (Table 1), while two ACC synthase (ACS1, MD01G1070400, and ACS6, MD14G1111500), one ACC oxidase (ACO3, MD09g1114800) and one ERF1 (MD05g1306900) encoding genes were down-regulated after three months of storage in wounded skins compared to control ones (Table 1 and Appendix A). Interestingly, a gene encoding the ethylene inducible alpha subunit of anthranilate synthase (ASA1), a rate-limiting step of tryptophan synthesis, was induced after 6 h of wounding, linking ethylene signalling with auxin homeostasis.

Several genes that were upregulated exclusively in the first two time points after wounding (6 h and 1 month), included genes involved in the regulation of jasmonic acid mediated signaling, in defence responses and salicylic acid catabolic processes, encoding for 2-oxoglutarate (2OG) and Fe(II)-dependent oxygenase superfamily proteins (MD11G1074400; MD03G1140400; MD14G1140400; MD04G1154400; MD14G1141200; MD03G1140700; MD14G1141000) (Table 1). Further down-regulation of salicylic acid responses could be evidenced after three months of storage (MD10G1111000) (Appendix A). Genes related to Jasmonic acid biosynthesis encoding a lypoxigenase (LOX)(MD09G1180900), an allene oxide cyclase (AOC)(MD16G1047500), and 12-oxophytodienoic acid reductases (OPRs) (MD15g1401100; MD15g1401300) were down-regulated after three months of storage, after the early transient up-regulation, six hours after wounding, of the same (MD16g1047500) or of different members (MD09G1084600, MD08G1038600) belonging to the allene oxide cyclase family. Interestingly, also genes encoding cold responsive fatty acid desaturases (FAD5 and FAD8, MD17G1212000 and MD04G1155400, respectively) as well as lipases (MD17G1265700, MD10G1282800, MD04G1049800, MD13G1187400) were induced in the early response to wounding (Appendix A).

Similar behaviours, concerning the transient up-regulation after 6 h and subsequent down-regulation after three months of storage in wounded peel tissues, could be identified for genes linked to ABA and IAA metabolism or response (Table 1 and Appendix A). A gene encoding the rate limiting ABA biosynthetic enzyme nine-cis-epoxycarotenoid dioxygenase 4, NCED4 (MD14G1105700), and a gene encoding an indole UDP-glucosyltransferase (UGT74E2) acting on conjugation of indole-3-butyric acid (IBA) (MD10G1111300) were up-regulated after 6 h from wounding while genes involved in ABA signalling (alpha subunit of heterotrimeric GTP-binding protein, MD15G1048800), response (ABA-responsive protein, MD14G1246200) or conjugation (UDP-glucosyl transferase, MD09g1141700) were down-regulated in wounded tissues after three months of storage. Interestingly, also transcription factors putatively involved in the enhancement of cold responses were transcriptionally induced as an early response to wounding: a gene encoding a C2H2 zinc finger family factor (MD03G1099300) (later down-regulated in wounded skins after three months of storage) and two genes encoding WRKY transcription factors (WRKY23 and WRKY72, MD17G1278100 and MD06G1189100). The former (WRKY23) was later down-regulated after three months of storage in wounded vs. control skins along with seven additional biotic stress and defence responsive WRKY encoding genes (MD15G1287300, MD01G1168600, MD04G1167700, MD00G1143500, MD17G1278100, MD17G1223100, MD05G1349800) (Appendix A). Noteworthy, seven genes encoding putative ubiquitin ligases (MD10G1021400, MD13G1017300, MD06G1232800, MD14G1040300, MD14G1239800, MD12G1040900, MD05G1020500) were specifically up-regulated in wounded apple peels after six hours.

## 3. Discussion

Apple superficial scald is a post-harvest chilling injury of great economic impact for sensitive apple varieties such as Granny Smith. The induction and development of scald symptoms have attracted interest from scientists since the origin of the apple cold storage industry [2,18] to understand its biochemical and molecular bases. Long periods of storage of apples at low temperatures, a standard applied protocol in the modern supply chain to extend the apple marketing period, result in the induction of this phenomenon. Superficial scald can be considered a chilling injury since early experiments have shown that intermittent warming of apples can reduce or prevent superficial scald incidence and it was soon understood that scald is an oxidative- and ethylene-dependent syndrome, since antioxidants (such as Diphenilamine or Etoxiquin) and inhibitors of ethylene biosynthesis (AVG) and/or action (1-MCP), respectively, can prevent it [2].

Several recent studies have contributed to clarify some biochemical and molecular fators putatively involved in either the induction of superficial scald or in the later development of symptoms during shelf-life. Different approaches have been exploited either to prevent or induce scald in a controlled way, in the search for the biochemical and molecular changes occurring during cold storage and, especially, during the process of scald induction by both metabolomics and transcriptomics approaches [8,9,10,18,67,68].

This manuscript focuses on adding more information on the molecular events that occur during cold storage that may play a regulatory role for scald initiation. We have built our experimental set-up on the basis of previous reports describing the preventive effect of wounding or mechanical damage of the apple skin on scald induction. We have chosen apple skin wounding as a specific perturbation tool enabling the local prevention of scald development. This choice was based on previous works reporting, by empirical observation, that superficial scald did not develop in the proximity of handling damages or of bitter pit lesions [2] and in the surrounding of artificially inflicted wounds performed by drawing a hypodermic syringe needle in the epidermis and sub-epidermal layers of scald susceptible Granny Smith apples [10]. We have first sought to confirm these results by applying an experimental set up which reproduced exactly that described by Abdallah et al. (1997). Indeed, our results confirmed during two consecutive years (2019 and 2020) that wounding applied only on one side of Granny Smith apples resulted in a slightly but significantly lower scald incidence in the surroundings of the wounded site in comparison to the non-wounded half of the same apple. To obtain a quantitative evaluation of scald incidence, a 2D image analysis of the entire apple surface was performed and the wounded and unwounded halves of the apples were independently analysed and compared for the extent of browned areas over the green intact ones. In this way we could reach a quantitative evaluation of the effect of wounding which resulted in nearly halved scald incidence in two consecutive vintages, that ranged from 20–30% of the surface in wounded skins compared to 30–60% detected in unwounded halves (Figure 1). The preventive effect of wounding, as expected, became most evident after three months of storage, after the chilling stress reached a certain threshold, and after seven days of shelf life. Our data overall provide a quantitative confirmation to the data reported by Abdallah et al. (1997) and confirm that wounding may exert a preventive action on scald induction.

To understand the signals underlying the scald inhibitory effect evoked by wounding we have performed an RNA-Seq characterisation of the global transcriptional profiles of the wounded versus non wounded halves of the apple fruits in a time course experiment. The evaluation of the transcriptional responses was carried out after 6 h from wounding, to identify the early molecular responses to the mechanical injury, and after one and three months of cold storage, the latter two time-points enabling the identification of the dynamics of the transcriptional regulation taking place during the scald inductive phase of storage. The peels of apples were sampled immediately after the exit from the storage chamber to obtain a picture of the events occurring during the inductive phase (storage in cold) and not during later development of symptoms (shelf-life).

Indeed, by comparing the wounding responsive transcriptional profile at each time point (statically) and by analysing the combined effect of wounding and of time of storage (dynamically) we could confirm that two transcriptional regulatory waves took place during storage.

An early transcriptional wave was induced soon after wounding (6 h) characterised by the prevailing up-regulation rather than by the down-regulation of genes. The finding that at three months of storage this effect was reversed, being the vast majority of DEGs down-regulated in wounded vs. healthy skins at this time-point, led us to hypothesise that the early wounding-associated events may prime a regulatory cascade finally leading to divergent gene expression dynamics during storage in the wounded halves of the fruits with respect to their unwounded counterparts. Such signalling cascade seemed to be limited to specific responses, since gene expression patterns were amply conserved between wounded and unwounded skins, showing that the overall processes of post-harvest ripening or senescence as well as responses to cold storage were amply maintained regardless of wounding. In fact, three groups of genes sharing a common expression pattern (either up- or down-regulation during cold storage) between wound and control skins were identified (Figure 4). Among these, genes responsible for ethylene signalling were included, specifically encoding ethylene receptors, as confirmed by the enrichment of the GO terms ethylene binding and ethylene receptor activity (Figure 4), confirming that at least some part of the ethylene signalling machinery, and thus of the post-harvest ripening process, remained unaffected by wounding. Therefore, on the basis of this evidence, the preventive effect of wounding on scald development should be hypothesised to rely on a defined subset of responses that are specifically and locally triggered by the mechanical injury. GO analysis of the transcripts up-regulated after 6 h in wounded tissues clearly pointed that such responses consisted in fact in the activation of transcriptional profiles enriched with genes that likely underlie the acquisition of resistance to stress, since the GO categories included responses to both biotic (responses to oomycetes, insects) and abiotic (wounding, hypoxia) stresses.

To pinpoint the signalling events subtending such responses and their consequences during storage, we have analysed the interactive effects between storage time and wounding on the RNA-Seq profiles, with the aim of pinpointing those gene sets whose expression dynamics were significantly influenced by the injury. By this approach we could confirm that wounding significantly changed the dynamics of the expression profiles for specific subsets of genes including altogether 448 DEGs. Indeed, these DEGs clearly pointed out the existence of different transcriptional waves. In fact, the genes that appeared to be transcriptionally up-regulated soon (6 h) after wounding later resulted in most cases to be down-regulated during the subsequent time points (Figure 5). Conversely, those DEGs that appeared to be up-regulated after three months of storage in control apple skin tissues were not regulated or even down-regulated during storage in wounded apple skins (Figure 5). The early wounding signalling effect resulted to be based to a certain degree on the regulation of hormonal responses, an aspect that became evident by both GO analyses and by a gene-by-gene study approach. An early response to wounding in apple skins appeared to be the up-regulation of genes encoding the jasmonic acid (JA) biosynthetic enzyme allene oxide cyclase (AOC) and of genes involved in fatty acid desaturation (FAD) and in lipid metabolism (lipases), a typical adaption response to cold stress for increased membrane fluidity and reduction of cold inducible disorders [3,11,69,70]. Both these processes are known to be related to wounding and cold responses, respectively, and to be involved in priming of the tissues against abiotic stress. Interestingly, the up-regulation of JA biosynthetic genes was followed by their down-regulation after three months in wounded skins in comparison to unwounded ones, for which an up-regulation was evident, suggesting that the transient early induction of JA may be responsible for triggering of the priming response in wounded tissues. It is interesting to note that a similar behaviour was evidenced for specific genes related to the biosynthesis and signalling of ethylene and of ABA. Ethylene is the prominent player not only in climacteric and apple fruit ripening [71] but also in the induction and development of scald [2]. In fact, the up-regulation 6 h after wounding of genes encoding two ERF transcription factors and one ACC oxidase (ACO3), followed by a later down-regulation of two main ripening ACC synthase (ACS1 and ACS6), one ACC oxidase (ACO3) and one ERF1 gene after three months of storage (Table 1, Appendix A), supports the hypothesis that wounding may induce transient co-ordinated ethylene and JA signalling waves. A concomitant transient wound-induced upraise of ABA biosynthesis and response may be also hypothesised on the base of the increased transcription of a gene encoding the NCED4 rate limiting biosynthetic enzyme. The increased transcription of transcription factors putatively involved in the enhancement of cold responses such as a C2H2 zinc finger family factor and two WRKY transcription factors (WRKY23 and WRKY72) in wounded apple tissues further confirms the activation of stress responses that may be downstream of the activation of such hormonal signals.

The role of JA biosynthesis and signalling in chilling stress tolerance is well known and supported by a large body of evidence. The biosynthesis and responses to JA are in fact upregulated in response to cold and to several biotic (including fungi and insect attacks) and abiotic stresses, and during senescence in several plant systems [69,70,72]. JA treatment increases tolerance to cold by inducing the degradation of the JA signalling repressors JAZMONATE ZIM-DOMAIN (JAZ) proteins JAZ1 and JAZ4, thus releasing their negative regulation on ICE1/2 and on CBF expression, the core transcriptional elements required for the activation of cold tolerance genes [73]. In fact, methyl jasmonate (MeJA) treatments have been reported by several authors to reduce cold induced injuries in fruits [74,75,76,77,78]. In peach the JA mediated alleviation of chilling stress injuries was attributed to the JA-induced stimulation of ethylene and sugar levels [74], suggesting that in this fruit system JA action may be mediated by ethylene.

Both synergistic and antagonistic effects have been reported for ethylene, ABA and JA in regulating resistance to pathogens, to abiotic stress (including cold) and to insect attack [72]. It may be speculated that the early up-regulation of JA, ethylene and ABA genes in wounded apple tissues could be connected, through a mechanism that will deserve further characterisation, with the prevention of a later climacteric build-up of transcripts of the same and of additional ethylene, JA and ABA genes that occurs during long term storage and that may be associated with senescence processes, with decay and with the induction of scald. JA plays in fact a dual positive role in the control of cold responses and in senescence [70].

Thus, the priming effect of wounding against scald development may rely on a concomitant and possibly synergistic signalling effect specifically evoked by wounding on JA, ethylene and ABA biosynthesis and signalling pathways. How precisely these signals may negatively regulate scald induction and development will need further studies.

As concluding remarks, it can be concluded that mechanical damage (wounding) causes a significant and early rearrangement of the hormonal biosynthetic and signalling balance in Granny Smith apple skins. Such changes result in a reversed hormonal landscape for ABA, JA and ethylene signals during later stages of post-harvest storage, which may explain the local inhibition of senescence and decay processes in wounded tissues. The transient hormonal reset induced locally by wounding in the immediate surroundings of the tissue injury will deserve further studies aimed at the characterisation of the signalling readout responsible for increased resistance of the apple epidermal and sub-epidermal cells to cold injury.

## 4. Materials and Methods

### 4.1. Experimental Setup, Apple Wounding and Storage Conditions

Apples Granny Smith (*Malus domestica* Borkh.) were harvested during two consecutive vintages in 2019 and 2020, at the experimental farm of the Edmund Mach Foundation (FEM) (San Michele all’Adige, TN, Italy) and held overnight at room temperature (20 °C). At harvesting the apples reached the physiologically mature stage and had an average starch index of 2.1 and 2.0 and an average firmness of 8.4 and 8.2 kg/cm^2^, respectively. A full list of the ripening parameters for the two vintages 2019 and 2020 is given in Table 2. 120 apples uniform in fruit size and colour and with absence of visible defects were randomly selected for each vintage. Fruits were subjected to wounding, as described below, and were placed on trays and then stored at 1.7 °C for up to 3 months in standard controlled atmosphere (CA): O_2_: 5%; CO_2_: 1.3%; Temperature 1.7 °C; UR: 60/70%. The progression of fruit ripening during storage was evaluated at each time point by measuring fruit firmness, sugars, acidity, starch and juiciness at the FEM facility by using the pimprenelle method (SETOP-Giraud Technologie, Cavaillon, France) (Appendix A).

Wounding was applied on one half of each fruit, keeping the opposite half of the fruit unwounded, as a control. Wounding was performed with a hypodermic needle (1 mm in diameter) by inserting it just under the skin parallel to the fruit surface starting from the fruit equator towards the blossom end. This region was chosen because scald symptoms are generally more severe in the subequatorial area with respect to the upper half of the fruit [10]. Eight wounds, with a distance of about 1 cm between each wound, were performed on one half of the apple taking care of not breaking the fruit skin over the wound, keeping the tissue damage at a subepidermal level. The single opening at each wound site was sealed with lanolin to prevent dehydration of the tissues, as also described by Abdallah AY et al. (1998), using a syringe to limit its application exclusively to the wounding site. The opposite side of each fruit was not wounded as a negative control (CTRL) (Figure 8).

### 4.2. Evaluation of Scald Incidence and Sampling of Apple Skin Material

Apples were sampled after six hours (T0), two weeks (T1), one month (T2) and three months (T3) of storage. At each sampling date, nine fruits per treatment were sampled to obtain three biological replicates, each made of a pool of three fruits. For the wounded half only the skin of the immediate surrounding of the wound area was excised taking a portion of the epidermis around the perforation site towards the blossom and for the non-wounded control half the skin was excised in the same area (yellow circles Figure 8) and all tissues were immediately frozen in liquid nitrogen for subsequent RNA extraction and further analyses. The remaining fruits (minimum number of 10 apples for vintage 2019 and 20 apples for 2020) were placed, at each time point, at room temperature for one week for the evaluation of scald development during shelf life. 2D images of the entire apple surface were taken, as described previously, at the day of sampling, after three days of shelf life and after 8 days of shelf life to track scald development. The percent of scald incidence was determined by image analyses on the whole apple surface. Panoramic 360° pictures of the fruits were taken by means of a prototype machine allowing the apple to rotate while acquiring the image. The image was then converted into an 8-bit file and black areas (scalded) were measured over the grey ones at each time point using Image J software (https://imagej.nih.gov/ij/ (last accession on 11 November 2021). For the statistical analysis the means of scald incidence of wounded and non-wounded apples were compared at each time point using *t*-test.

### 4.3. RNA Extraction and Expression Analysis

All expression analyses were carried out on samples collected in the 2019 vintage at 6 h, 1 month and three months of storage after wounding. The two-week time point was excluded from subsequent analyses since no changes in marker genes and physiological parameters were observed and on the base of our previous observations [9]. Total RNA was isolated from approximately 500 mg of apple skin tissue grinded in liquid nitrogen as described by Zermiani et al. (2015) [9]. Total RNA was resuspended in 50 µL of nuclease free water. Subsequently, DNAse digestion was performed with RQ1 RNAse-free DNAse (Promega, Milano, Italy). Total RNA quality and integrity were checked on a Bioanalyzer 2100 (Agilent, Santa Clara, CA, USA), for all samples a RIN (RNA integrity number) ≥ 8.0 was detected. cDNA synthesis was performed following Takara PrimeScript IV 1st strand cDNA Synthesis guidelines, starting form 0.5 µg of total RNA, as described by Nonis et al., 2012 [79].

Homogeneous samples have been identified through quantitative Real-Time PCR expression analysis on known ripening marker genes (ACO and PPO) using a StepOnePlus™ Real-Time PCR System (Applied Biosystems, Thermo Fisher Scientific, Waltham, MA, USA) and the FAST SYBR^®^ GREEN PCR Master Mix (Thermo Fisher Scientific, Waltham, MA, USA), following the manufacturer’s guidelines. Melting curves analysis revealed a single amplification product in each reaction. Three technical replicates were carried out for each primer combination in each sample and a relative quantification of gene expression (normalized to MdCOST8283 transcript quantities) was performed with the StepOne Software 2.3 (Thermo Fisher Scientific, Waltham, MA, USA). Primer sequences of ripening marker genes targets (ACO and PPO) and of DEGs identified through RNA-Seq analyses are reported in Table 3.

RNA Sequencing was performed at the Centro di Ricerca Interdipartimentale per le Biotecnologie Innovative (CRIBI, Padova, Italy), on a NextSeq500 (Illumina, San Diego, CA, USA) instrument on three selected time points (T0 = 6 h, T2 = 1 month and T3 = 3 months after wounding, respectively) on three independent biological replicates for each time point and experimental condition.

### 4.4. RNA-Seq Analysis and Data Processing

Illumina directional sequencing of mRNA was performed at Centro di Ricerca Interdipartimentale per le Biotecnologie Innovative (CRIBI—University of Padova, Italy) on a NovaSeq 6000 instrument (Illumina, San Diego, CA, USA). For each sample x replicate combination, 25–35 M paired-end reads of 150 nucleotides were generated. The quality of reads was assessed using FastQC (https://www.bioinformatics.babraham.ac.uk/projects/fastqc/ (last accession on 30 March 2021)). The sequenced reads were pre-processed for low quality sequence filter and adapter trimming with with ERNEFILTER 2.1.1 (Del Fabbro et al., 2013), and Trimmomatic (Bolger et al., 2014), respectively. Hight quality reads were mapped to the Malus x domestica GDDH13 v1.1 genome obtained from the Rosaceae database (Malus x domestica GDDH13 v1.1 Whole Genome Assembly and Annotation retrieved from GDR database (www.rosaceae.org) (last accession on 30 March 2021); Daccord N et al., 2017) using the spliced aligner HISAT2 [80]. Gene expression counts were generate using featureCounts software program [81]. The differential expression analysis was carried out using DESeq2 (Love et al., 2014) applying both the default Wald-test for pairwise comparisons and the likelihood ratio test (LRT) for time-course analyses. Gene Ontology (GO) enrichment analysis was performed using g:Profiler (https://biit.cs.ut.ee/gprofiler) (last accession on 14 April 2021)) after GO functional annotation of apple transcripts using Trinotate [82].

## Figures and Tables

**Figure 1 ijms-22-13425-f001:**
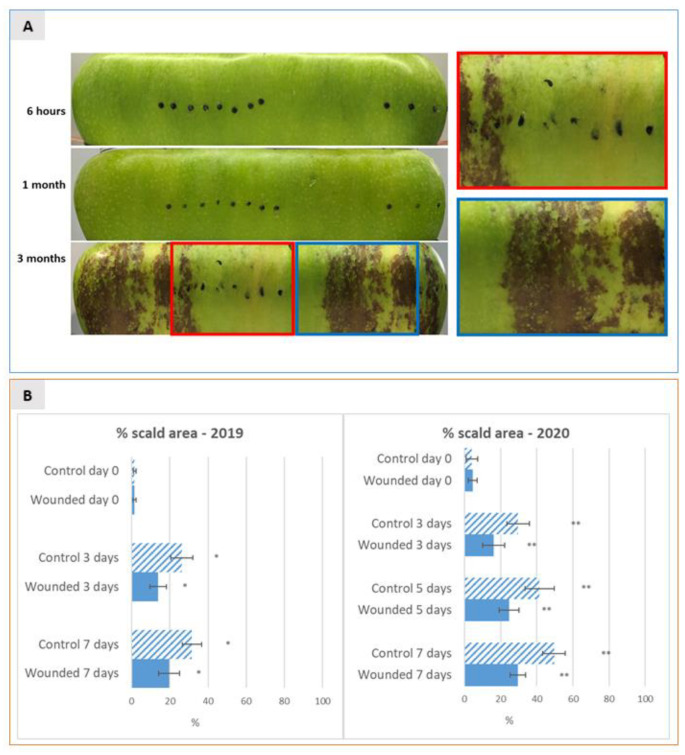
(**A**) 2D pictures of the entire surface of Granny Smith apples at six hours after wounding (before storage) and after one and three months of cold storage (1.7 °C) followed by 8 days of shelf life. Pictures were taken at 3, 5 and 8 days of shelf-life by two-dimensional acquisition of the entire apple surface as described in materials and methods. Representative pictures are shown at the 8th day of shelf life, when symptoms of scald development became clearly visible. Black dots were applied on the apple surface to highlight the exact sites where wounding was performed. The right panel in A shows an enlargement of the boxed insets, to give a close-up view of the difference in terms of scald development between the wounded (red box) and the unwounded (blue box) areas of the apple surface after three months of storage followed by 8 days of shelf-life. (**B**) Dynamics of superficial scald incidence were expressed as mean percentage (±SD) of browned scald area over the total apple surface, measured on a total number of 10 (2019) and 20 (2020) independent apples (replicates) for each thesis after three months of cold storage and evaluated by 2D acquisition at different time-points of shelf-life over a period of 8 days (at 0, 3 and 8 days in 2019 and at 0, 3, 5 and 8 days in 2020). The incidence is shown separately for the wounded apple half versus non-wounded control half of apples. * and ** indicate a statistically significant difference between the means with a *p*-value of 0.1 and 0.05, respectively.

**Figure 2 ijms-22-13425-f002:**
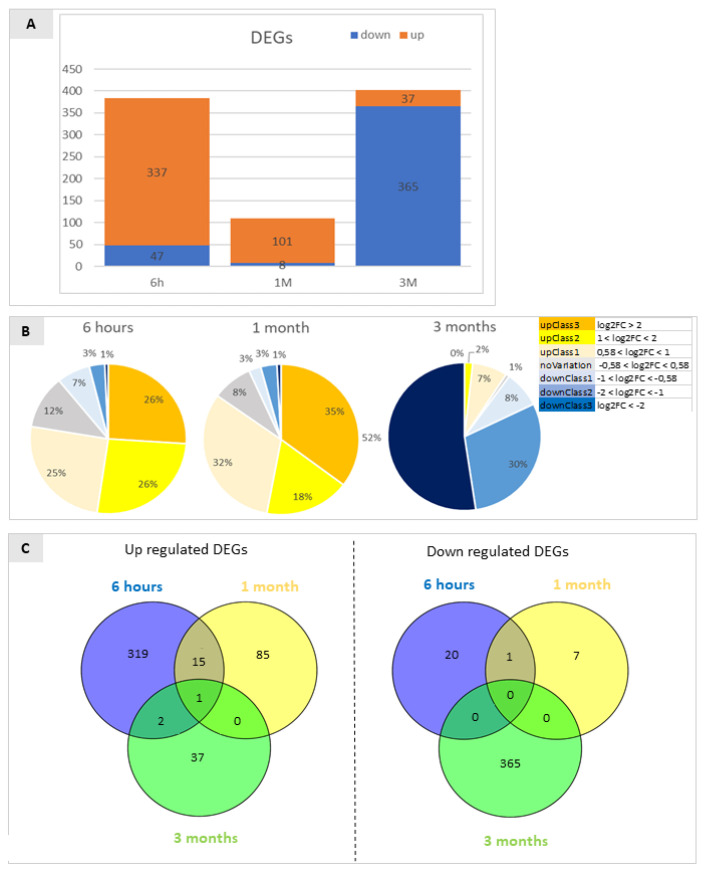
(**A**) Diagram representing the percentage of genes identified as differentially expressed by RNA-Seq analyses in the comparison between wounded and control (unwounded) Granny Smith apple skins at 6 h after wounding (6 h), and after one and three months of storage (1 M and 3 M). Only DEGs with FDR < 5% were included. DEGs were classified as up-regulated or down-regulated (comparison between the treatments is >1 or <1, respectively, accordingly to their FPKM values. The prevalence of mainly up-regulated DEGs in the first two time-points is reversed at the last time point when down-regulated DEGs prevailed. (**B**) Fold change distribution of the DEGs identified by the RNA-Seq experiments in the three comparisons analysed. The ratios reported represent the comparison of wounded vs. control at the three time points of interest. Orange, yellow and pale yellow represent the fraction of genes that are up regulated, the grey area indicates the percentage of genes that show no statistically significant variation, and the blue fractions indicate the percentage of genes that are down regulated in wounded tissue in respect to the control. Different shades indicate the entity of variation based on the log2FC. (**C**) Venn diagrams showing the repartition of down-regulated DEGs and up-regulated DEGs in the comparison between wounded and control samples at three time points. Non-overlapping numbers represent the number of genes unique to each treatment. Overlapping numbers represent the number of mutual genes between treatments. The data refer to samples collected in 2019.

**Figure 3 ijms-22-13425-f003:**
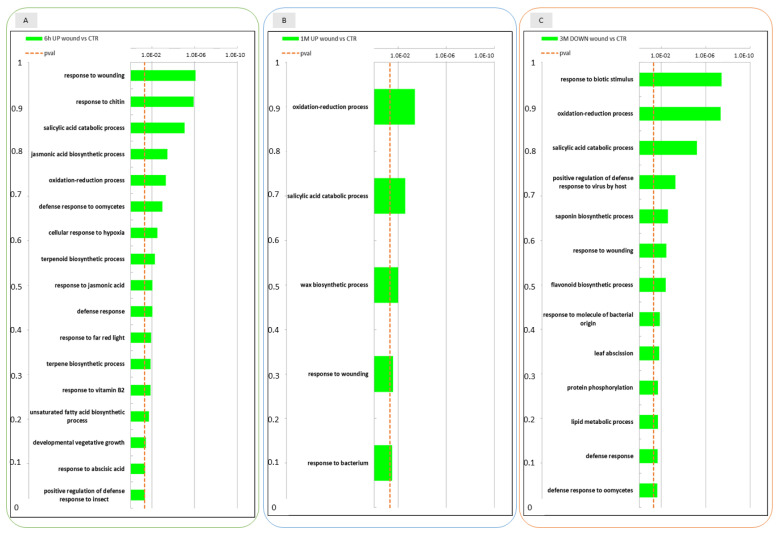
Enrichment analysis of DEGs using the Gene Ontology database. GO categories overrepresented among up- and down-regulated gene sets in the wounded apple skin in comparison to non-wounded control skin at three time points 6 h after wounding (6 h), 1 month (1 M) and 3 months (3 M) of conservation. The data refer to samples collected in 2019. In panel (**A**) GO categories are shown for genes that were upregulated in the comparison between wound and control at 6 h; panel (**B**): GO categories of genes upregulated in the comparison between wound and control at 1 M; panel (**C**): GO categories of genes that were downregulated in the comparison between wound and control at 3 M.

**Figure 4 ijms-22-13425-f004:**
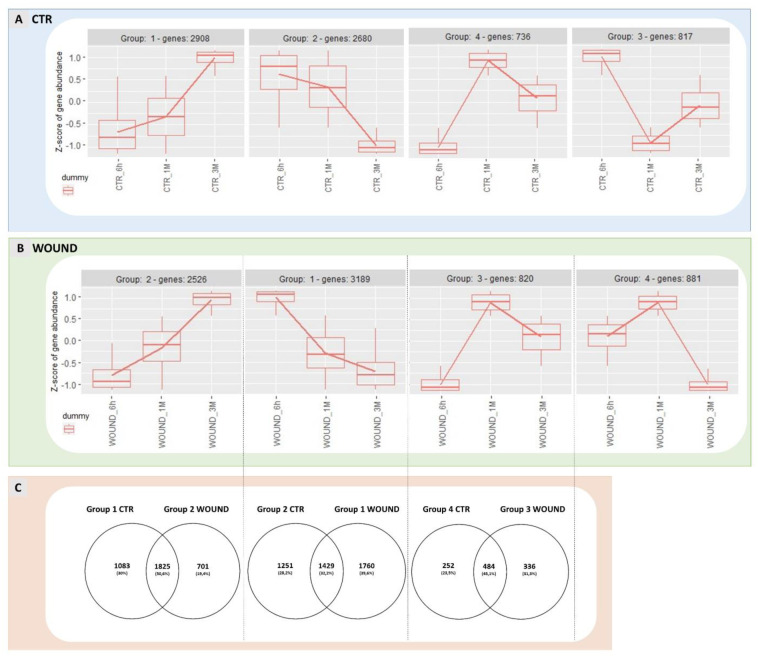
Expression patterns identified for DEGs regulated in control apples over time are shown in panel (**A**) while expression patterns of wounded apples are displayed in panel (**B**). Analogue gene expression patterns for wounded and control apples are shown one on top of another in comparison and Venn Diagrams show the number of genes shared by the two treatments or exclusive for wounded or control apples (panel (**C**)). Panel (**D**) shows the GO analysis of genes that were found shared between analogue expression groups in control and wounded apples. The data refer to samples collected in 2019.

**Figure 5 ijms-22-13425-f005:**
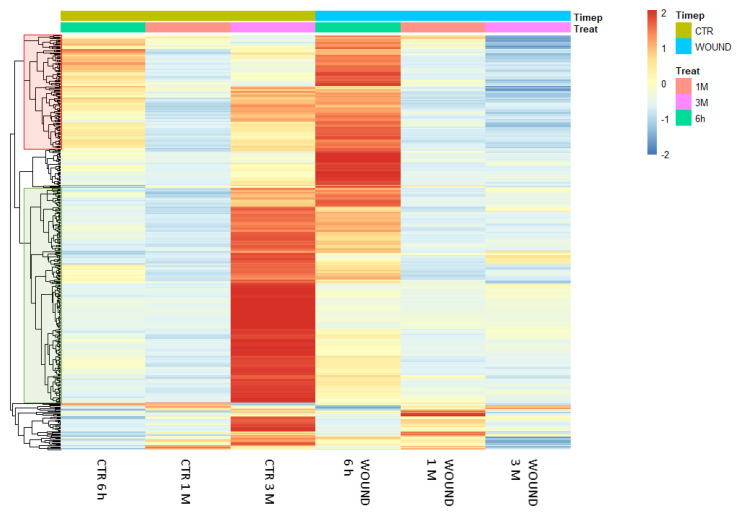
Heatmap showing the expression patterns of 448 differentially expressed genes in the treatment x time interaction analysis in wounded and control tissues of Granny Smith apples. Red and blue colours represent an increase or decrease in the gene expression levels, respectively. DEGs were identified as significantly differentially expressed with an adjusted *p*-value of <0.01 out of 28313 genes with nonzero total read counts using DESeq2. The data refer to samples collected in 2019.

**Figure 6 ijms-22-13425-f006:**
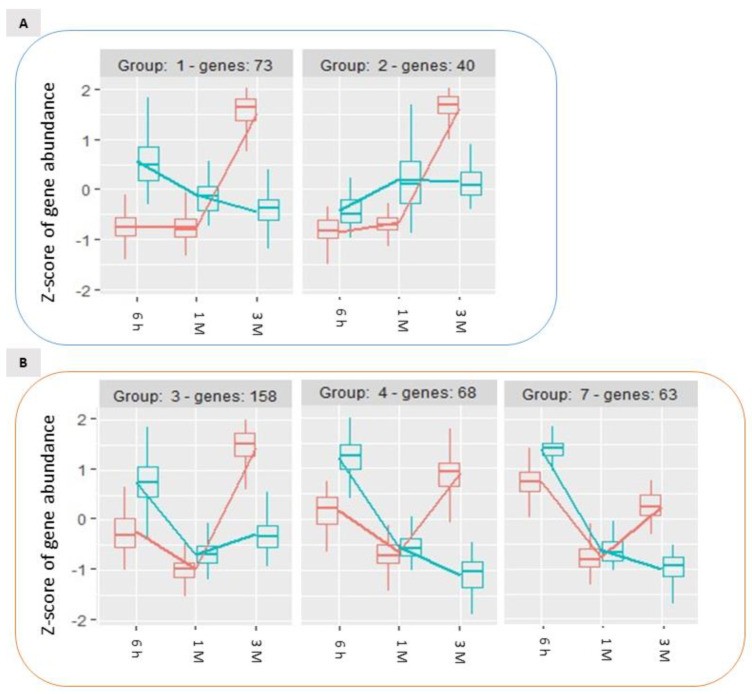
Time-course expression profiles identified by the time per treatment analysis among storage-regulated DEGs in control and wounded apples, highlighting the significantly different temporal dynamics of expression of specific groups of genes ((**A**–**C**); see text for details) as a consequence of wounding. The data refer to samples collected in 2019.

**Figure 7 ijms-22-13425-f007:**
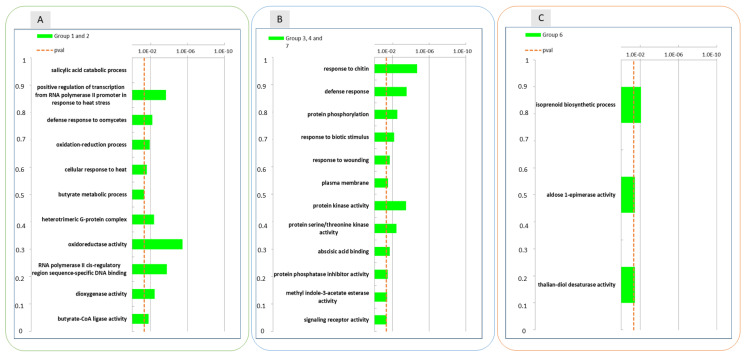
Enrichment analysis of DEGs with similar gene expression patterns identified in the treatment and time interaction using the Gene Ontology database between the groups of DEGs reported in Figure 6: groups 1 and 2 (Panel (**A**)), groups 3,4 and 7 (Panel (**B**)) and group 6 (Panel (**C**)). The data refer to samples collected in 2019.

**Figure 8 ijms-22-13425-f008:**
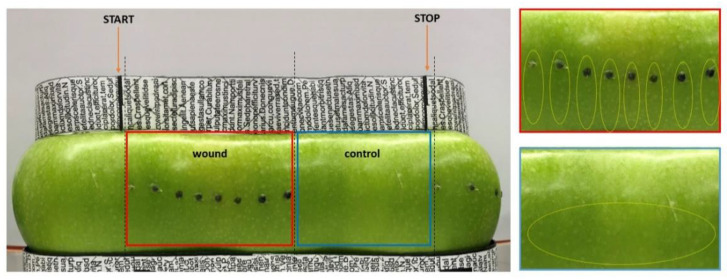
Experimental set-up for the 2D image acquisition of the whole apple surface area and for sampling of material. Wounding sites are evidenced by black spots to allow tracking. The picture was taken by acquiring the entire surface of the fruit by placing it onto a rotating platform and using reference points (evidenced by black boxes and lines located on the adaptor placed on top and bottom sides of the apple), enabling to point the start and end of the image acquisition, as evidenced by orange arrows. By using the reference points the wounded and unwounded (control) halves of the fruits could be precisely separated (evidenced by the boxed areas). This approach allowed acquisition of almost the entire fruit’s surface and for later analysis of the percentage of browned (scalded) versus green areas. On the right of the panel enlarged pictures of the wounded (red box) and of the non-wounded control (blue box) area are shown. The yellow dotted circles indicate the skin area that has been excised for sampling.

**Table 1 ijms-22-13425-t001:** Regulatory genes differentially expressed after 6 h of wounding and after 1 and 3 months of cold storage (only DEGs with a log2FC > |3.5| are listed). Columns from left to right report: description of the encoded protein, the corresponding *Rosaceae* database ID of the encoding gene, the closest *Arabidopsis* homologue, log2 fold change of transcript abundance found between wounded and control samples, putative functional/regulatory process played by the closest *Arabidopsis* homologue, and corresponding reference. The data refer to samples collected in 2019.

**DEGs log2FC > 3.5–6 h**					
**Description**	**GGDH 13 v1.1**	**Arabidopsis Homologue**	**log2FC**	**Function/Regulatory Process**	**References**
2-oxoglutarate (2OG) and Fe(II)-dependent oxygenase superfamily protein	MD14G1140400	AT4G10490.1	9.1359	defense response to oomycetes, salicylic acid catabolic process	[15]
2-oxoglutarate (2OG) and Fe(II)-dependent oxygenase superfamily protein	MD04G1154400	AT3G11180.1	8.8233	regulation of jasmonic acid mediated signaling pathway	[16]
2-oxoglutarate (2OG) and Fe(II)-dependent oxygenase superfamily protein	MD14G1141200	AT4G10490.1	8.6543	defense response to oomycetes, salicylic acid catabolic process	[15]
cytochrome P450%2C family 94%2C subfamily C%2C polypeptide 1	MD03G1140700	AT2G27690.1	7.0295	response to wounding	[17]
2-oxoglutarate (2OG) and Fe(II)-dependent oxygenase superfamily protein	MD14G1141000	AT4G10490.1	6.8804	defense response to oomycetes, salicylic acid catabolic process	[15]
potassium transporter 1	MD05G1223100	AT2G30070.1	5.8179	potassium ion transmembrane transport	[18]
laccase 7	MD12G1157100	AT3G09220.1	5.0617	member of laccase family of genes (17 members in Arabidopsis)	[19]
Integrase-type DNA-binding superfamily protein	MD10G1032000	AT5G64750.1	4.9703	wounding stress response	[20]
UDP-glucosyl transferase 85A2	MD08G1185700	AT1G22360.1	4.7899	glucuronosyltransferase activity	
Phosphorylase superfamily protein	MD15G1191300	AT4G28940.1	4.7704	nucleoside metabolic process	
hypothetical protein	MD06G1215300	AT1G13360.1	4.7629	cellular response to hypoxia	
high affinity K+ transporter 5	MD11G1303100	AT4G13420.1	4.5891	potassium ion transport	[21]
terpene synthase 14	MD10G1309900	AT1G61680.1	4.5332	terpene synthase	[22]
Uncharacterized protein family (UPF0114)	MD00G1038700	AT4G19390.1	4.4614	Na+ efflux activity	[23]
F-box family protein	MD11G1169800	AT2G27310.1	4.4381	converts 2,3-oxidosqualene to cycloartenol in the sterol biosynthesis pathway	[24]
RING/U-box superfamily protein	MD09G1240700	AT2G42360.1	4.3325	protein ubiquitination	
basic helix-loop-helix (bHLH) DNA-binding superfamily protein	MD15G1305200	AT4G37850.1	4.3299	regulation of transcription	
plastid movement impaired protein	MD16G1029200	AT2G01340.1	4.2995	response to nematode	[25]
AZA-guanine resistant1	MD07G1231600	AT3G10960.1	4.2822	adenine transport, guanine transport	[26]
cytochrome P450%2C family 82%2C subfamily G%2C polypeptide 1	MD10G1158100	AT3G25180.1	4.0730	defense response to insect wounding	[27]
anthranilate synthase alpha subunit 1	MD07G1216200	AT5G05730.1	4.0691	response to wounding	[28]
Protein kinase superfamily protein	MD15G1076300	AT4G00340.1	4.0232	Ser/Thr receptor-like protein kinase	[29]
RING-H2 finger protein 2B	MD13G1051400	AT2G01150.1	3.8958	ABA response to abiotic stress/positive regulation of abscisic acid-activated signaling pathway	[30]
myb domain protein 74	MD05G1327500	AT4G05100.1	3.8391	response to salt stress	[31]
MLP-like protein 423	MD13G1161700	AT1G24020.2	3.8376	plant development/defence response	
polyamine oxidase 1	MD02G1306200	AT5G13700.1	3.7689	reducing Reactive Oxygen Species Production and Increasing Defense Gene Expression	[32]
Leucine-rich repeat protein kinase family protein	MD06G1198600	AT1G74360.1	3.6803	defense response to nematode	[33]
Calcium-binding EF-hand family protein	MD03G1144700	AT2G27480.1	3.6779	calcium-binding EF-hand family protein	
WRKY DNA-binding protein 72	MD06G1189100	AT5G15130.1	3.6274	transcriptional reprogramming associated with plant immune responses	[34]
ARM repeat superfamily protein	MD06G1232800	AT5G37490.1	3.5579	plant-type hypersensitive response	[35]
NAD(P)-binding Rossmann-fold superfamily protein	MD06G1003700	AT2G33590.1	3.5458	response to ABA	[36]
myb domain protein 15	MD05G1197600	AT3G23250.1	3.5358	response to cold	[37]
cytochrome P450%2C family 94%2C subfamily C%2C polypeptide 1	MD11G1171100	AT2G27690.1	3.5234	response to wound	[17]
cytochrome P450%2C family 71%2C subfamily B%2C polypeptide 37	MD16G1103600	AT3G26330.1	3.5031	oxidoreductase activity	
**DEGs log2FC > 3.5–1 month**					
**Description**	**GGDH 13 v1.1**	**Arabidopsis Homologue**	**log2FC**	**Function/Regulatory Process**	**References**
Glucose-methanol-choline (GMC) oxidoreductase family protein	MD03G1090900	AT1G73050.1	7.2112	drought resistance	[38]
2-oxoglutarate (2OG) and Fe(II)-dependent oxygenase superfamily protein	MD11G1074400	AT1G52800.1	7.1643	hormonal development regulation	
Glucose-methanol-choline (GMC) oxidoreductase family protein	MD00G1012900	AT1G73050.1	5.6504	drought resistance	[38]
RmlC-like cupins superfamily protein	MD10G1022900	AT5G38940.1	5.0327	response to abscisic acid (ROP10)	
ethylene response factor 1	MD16G1216900	AT3G23240.1	4.7101	ethylene-activated defence response to abiotic factors	
2-oxoglutarate (2OG) and Fe(II)-dependent oxygenase superfamily protein	MD03G1140400	AT4G10490.1	4.6868	defense response to oomycetes, response to salicylic acid	[15]
nitrate transporter 1.5	MD02G1228800	AT1G32450.1	4.4253	response to nitrate/nitrate transport	[39]
Pathogenesis-related thaumatin superfamily protein	MD17G1250000	AT1G20030.2	4.4096	oxidative stress response	[40]
Eukaryotic aspartyl protease family protein	MD11G1287900	AT1G03220.1	4.4009	response to abiotic stress conditions	[41]
Pathogenesis-related thaumatin superfamily protein	MD17G1249600	AT1G20030.2	4.0734	oxidative stress response	[40]
Late embryogenesis abundant (LEA) hydroxyproline-rich glycoprotein family	MD06G1094100	AT2G46150.1	3.944	pathogen response	[42]
expansin A1	MD06G1195100	AT1G69530.2	3.9284	regulation of stomatal opening	[43]
homolog of carrot EP3-3 chitinase	MD04G1047700	AT3G54420.1	3.8788	response to environmental stresses (cold, wounding, dehydration)	
expansin A4	MD01G1135600	AT2G39700.1	3.8372	growth and development modulation	[44]
cytochrome P450%2C family 72%2C subfamily A%2C polypeptide 9	MD16G1056600	AT3G14630.1	3.6968	hydrolyzation of gibberellins	[45]
**DEGs log2FC < 3.5–3 months**					
**Description**	**GGDH 13 v1.1**	**Arabidopsis Homologue**	**log2FC**	**Function/Regulatory Process**	**References**
UDP-Glycosyltransferase superfamily protein	MD17G1125800	AT1G22400.1	−5.0363	plant growth and development	[46]
Plant invertase/pectin methylesterase inhibitor superfamily	MD02G1207900	AT2G45220.1	−4.8852	pathogen-induced pectin methylesterases activity and resistance against B. cinerea by triggering jasmonic acid–ethylene-dependent PDF1.2 (AT5G44420) expression	[47]
ACC synthase 1	MD01G1070400	AT3G61510.1	−4.4324	ethylene biosynthetic process	[48]
Cupredoxin superfamily protein	MD15G1328900	AT4G39830.1	−4.2877	salt-stress tolerance	[49]
NC domain-containing protein-like protein	MD01G1232800	AT5G16360.1	−4.1931	unknown	
NAD(P)-linked oxidoreductase superfamily protein	MD12G1240300	AT1G60690.1	−4.1175	unknown	
blue-copper-binding protein	MD02G1028300	AT5G20230.1	−4.1059	response to cold/freezing	[50]
cinnamyl alcohol dehydrogenase 7	MD01G1042500	AT4G37980.1	−4.1012	direct plant defense at wound sites	[51]
1-aminocyclopropane-1-carboxylic acid (acc) synthase 6	MD14G1111500	AT4G11280.1	−4.0773	environmental stress responses	[52]
phospholipase A 2A	MD15G1085800	AT2G26560.1	−4.0524	response to dehydration	[53]
ABC-2 and Plant PDR ABC-type transporter family protein	MD09G1204300	AT1G59870.1	−3.9073	ccadmium stress tolerance	[54]
Late embryogenesis abundant (LEA) hydroxyproline-rich glycoprotein family	MD07G1281600	AT2G46150.1	−3.8798	pathogen response	[42]
potassium transporter 1	MD05G1223100	AT2G30070.1	−3.8791	response to salt stress	[55]
Late embryogenesis abundant (LEA) hydroxyproline-rich glycoprotein family	MD14G1114000	AT2G46150.1	−3.8079	pathogen response	[42]
NAD(P)-linked oxidoreductase superfamily protein	MD08G1240600	AT1G60750.1	−3.8033	response to wounding	[56]
S-adenosyl-L-methionine-dependent methyltransferases superfamily protein	MD08G1242800	AT5G10830.1	−3.7399	ROS-induced cell death	[57]
cytochrome P450%2C family 81%2C subfamily D%2C polypeptide 8	MD03G1281500	AT4G37370.1	−3.6903	response to heavy metal	[58]
FAD-binding Berberine family protein	MD10G1244300	AT4G20820.1	−3.685	oxidative stress response	[59]
early nodulin-like protein 14	MD02G1028800	AT2G25060.1	−3.6798	male-female communication and fertilization	[60]
UDP-glucosyl transferase 73B3	MD07G1007400	AT4G34131.1	−3.6701	regulation of redox status and general detoxification of ROS-reactive secondary metabolites	[61]
SAUR-like auxin-responsive protein family	MD07G1117400	AT3G60690.1	−3.6455	Auxin-responsive protein SAUR like	
calcium ATPase 2	MD02G1185000	AT4G37640.1	−3.6231	Ca^2+^ transport against concentration gradients using ATP	[62]
Heavy metal transport/detoxification superfamily protein	MD10G1280500	AT1G06330.1	−3.5779	response to heavy metal	[63]
cellulose synthase like G1	MD02G1095700	AT4G24010.1	−3.5496	response to cold	[64]
multidrug resistance-associated protein 3	MD16G1109500	AT3G13080.1	−3.5382	response to heavy metal	[65]
blue-copper-binding protein	MD02G1028600	AT5G20230.1	−3.5213	ROS and SA-synthesis-related genes expression regulation	[66]

**Table 2 ijms-22-13425-t002:** Mean fresh weight and ripening parameters (starch content, sugar content (brix°), firmness and acidity) measured at harvest in the two experimental vintages 2019 and 2020 on the apples that have been used for the experiments.

	Mean Starch Content	Mean Fresh Weight	Sugars Brix	Firmness kg/cm^2^	Acidity g/L mal.ac
vintage 2019	2.1	212.2	9.7	8.4	9
vintage 2020	2.0	189	10	8.1	7.9

**Table 3 ijms-22-13425-t003:** Primer sequences used for targeted Real-Time PCR expression analysis on *MdACO* and *MdPPO*, known as ripening marker genes, and of RNA-Seq identified DEGs. ACO, 1-aminocyclopropane-1-carboxylate oxidase—ACC oxidase; PPO, Polyphenol oxidase; BCB, blue-copper-binding protein; EP3, Endochitinase EP3; 2OG, 2-oxoglutarate (2OG) and Fe(II)-dependent oxygenase superfamily protein; FAD7, fatty acid desaturase 7; MLP423, Major latex-like protein 423, LHCB6, light harvesting complex photosystem II subunit 6; LHCB4.2, light harvesting complex photosystem II subunit 4.2.

Gene Bank Accession Number	Name	Primer fw	Primer rv	References
AB030859	MdACO	CAGTCGGATGGGACCAGAA	GCTTGGAATTTCAGGCCAGA	Dal Cin et al., 2000
L29450	MdPPO	CTGACTCGGACTGGTTGGAC	CTTCGCTACTTTGCTCAATGC	this work
MD02G1028600	BCB	TCCTCCAGCTGGCTCTGTTGC	GCTACTCTGTGTTCCCCGCTCC	this work
MD04G1047700	EP3	TGGCCTCCGATCCCGTCCTT	AGGGCAGGCTGTTTACCATCACA	this work
MD11G1074400	2OG	CCTCACACGGACAAGAATTTCACA	GCAGCCTCAATCCACTCGCC	this work
MD10G1299400	unknown1	CCACTGCCCGCGATTTACG	ACCAGAGTCGGGGGCTGATG	this work
MD11G1072300	FAD7	AGCAGCAGAGCAAGCCCATCAA	GGAGGGGCACTTGGGTCGAA	this work
MD13G1161200	unknown2	GAGCACCAGCCACTATCACACCAAG	GGTGGGAGGCCTTCTCTTTGC	this work
MD13G1161000	unknown3	TGGTGGCTTCCGGCAATGGT	GGTGGGAGGCCTTCTCTTTGC	this work
MD13G1160900	MLP423	CAGTCTGCGCTCCTGCTAGGTTG	TTCCAACACCTCCATCTCCCTCA	this work
MD15G1411600	LHCB6	GGCTCGATGGCTCGCTTCC	ACCACTTGAGGAACGCCGGG	this work
MD04G1151300	LHCB4.2	CCCCGTCGCACCTTAGCACA	GTAGGCGTCGGTCGGAAGGC	this work

## Data Availability

The data that support the findings of this study are available from the corresponding author upon reasonable request.

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
