# Peer review of "Transcriptomic Insights on the Preventive Action of Apple (cv Granny Smith) Skin Wounding on Superficial Scald Development"

_ijms, 2021, doi:10.3390/ijms222413425_

Round 1
Reviewer 1 Report
Dear Editor, in the manuscript IJMS-1484356 authors evaluated the effect of wounding the epidermal and sub-epidermal layers of apple skins on the prevention of superficial scald in the surroundings of the wounds during apple storage under controlled atmosphere in two independent vintages. In general, the experiments were well performed and the manuscript is well written providing new and interesting information. Time course RNA-seq analyses of the transcriptional changes in wounded versus un wounded skins revealed that two transcriptional waves occurred, an early wave included genes up-regulated by wounding already after 6 hours, these genes being connected to the biosynthesis and signalling of JA, ethylene and ABA while the second wave occurred after three months of storage and thee genes were up-regulated exclusively in unwounded skins and prevented from its occurrence in wounded skins. Thus, the manuscript could be suitable for publication, although the following comments should be considered:
- Why were samples taken after 6h, 1 and 3 months? Was it based on previous results? This issue should be clarified in the revised manuscript.
- Figure 1 legend: Add information about if data are the mean ± SD and the number of replicates.
- Figure 1b and figures 3, 4, 6 are too small.
- Letters A, B, C and D should be added to figures 6.
- Line 309: Figure 4, B should be cited.
- References on Table 1 should be cited by using numbers and all of them should be added to reference list because most of them are missing.
- Check result section and move paragraph with references to Discussion section.
- In figures 2, 3, 4, 5, 6 and 7 it should be indicated if data are from 2019 or 2020 experiments.
- Line 683: Please clarify because data of samples of 2 weeks of storage are nor provid3d.
- Check the whole manuscript and cite references by using number, according to the journal format.
Author Response
We thank the reviewer for his/her helpful suggestions and comments. We hope we have addressed the points raised by the reviewer. Please find our responses to each comment as reported below:
Reviewer comment 1: - Why were samples taken after 6h, 1 and 3 months? Was it based on previous results? This issue should be clarified in the revised manuscript.
Answer 1: We have added the following sentences to explain the choice of the time-points for RNA-Seq and molecular analyses in the results section on page 3, based on published works by other laboratories (reviewed by Lurie and Watkins, 2012) and by our own lab (Zermiani et al., 2015): “The choice of the time point at six hours was made to highlight the early regulatory transcriptional responses to wounding. The time points after one and three months of storage were chosen to highlight the medium and long term transcriptional regulations taking place before and after the induction of superficial scald, respectively. This choice was based on previously published works showing that the cold stress threshold for superficial scald induction is commonly reached by Granny smith apples after a minimum of two to three months of cold storage (Lurie and Watkins, 2012; Zermiani et al., 2015).” We hope that this has clarified our choice of the time-points.
Reviewer comment 2: - Figure 1 legend: Add information about if data are the mean ± SD and the number of replicates.
Answer 2: the data were replicated individually on 10 and 20 independent fruits in 2019 and 2020, respectively, for each time point and experimental condition. This information is now included explicitly in the legend of Figure 1, as requested by the reviewer, and is also mentioned in the M&M at page 25 (paragraph on “Evaluation of scald incidence”, lines 690-691).
Reviewer comment 3: - Figure 1b and figures 3, 4, 6 are too small.
Answer 3: the size of the figures and of font has been increased, where possible, to make them more readable. In some cases (such as GO analyses) the font is defined by the programme used. We have anyway included bigger sized figures in the manuscript. We have also changed Figure 5 by removing the ID of the genes that were all collapsed, were unnecessary and appeared like black stripes on the right hand side of the picture. We hope that the new Figures are now satisfactory.
Reviewer comment 4: - Letters A, B, C and D should be added to figures 6.
Answer 4: addressed. Letters have been added as requested
Reviewer comment 5: - Line 309: Figure 4, B should be cited.
Answer 5: The Figure 4,B is now cited in the main text, according to the reviewer’s suggestion
Reviewer comment 6: - References on Table 1 should be cited by using numbers and all of them should be added to the reference list because most of them are missing.
Answer 6: we apologize very much for this rather big mistake. We have modified the references in Table 1 by numbering them and added them to the Reference list, as requested.
Reviewer comment 7: - Check result section and move paragraph with references to Discussion section.
Answer 7: we have removed the sentence with citations from the results section on page 20 (lines 450-452 of the original ms) and we have simplified the text accordingly by simply describing the results obtained and removing comments that may recall a discussion. We hope we have correctly interpreted the reviewer’s suggestion/request.
Reviewer comment 8: - In figures 2, 3, 4, 5, 6 and 7 it should be indicated if data are from 2019 or 2020 experiments.
Answer 8: we have addressed this point raised by the reviewer by adding the sentence “The data refer to samples collected in 2019” at the bottom of the legends of Figures 2 to 7. We have also added a sentence in the M&M section, paragraph “RNA extraction and Expression analysis” (Page 27 of the revised manuscript) stating that “All expression analyses were carried out on samples collected in the 2019 vintage”, since Figures 2-7 refer to these RNA-Seq analyses.
Reviewer comment 9: - Line 683: Please clarify because data of samples of 2 weeks of storage are nor provid3d.
Answer 9: we have added this information in the M&M section, paragraph 4.3 “Rna extraction….” on page 27. The two weeks time point was initially sampled and analysed but, as we also previously experienced (Zermiani et al, 2015), at this time point no significant events could be evidenced nor for the marker genes nor for physiological parameters and therefore was excluded from the subsequent molecular analyses, also to keep the RNA-Seq costs within an acceptable range. We thus kept the focus on the early (6 hours), intermediate (1M) and late (3M) responses. We hope we have addressed the reviewer’s request by adding this explanatory sentence in this section of the M&M.
Reviewer comment 10: - Check the whole manuscript and cite references by using number, according to the journal format.
Answer 10: We have carefully double checked the numbering of references and their correct citation in the text, as requested.
Reviewer 2 Report
The manuscript ijms-1484356 provides wide and accurate transcriptomic research on the preventive action of apple skin wounding on superficial scald development. I congratulate the authors for the present results and conclusions that are of great interest to researchers and industry. The authors concluded that mechanical damage (wounding) causes a significant and early rearrangement of the hormonal biosynthetic and signalling balance in Granny smith apple skins.
I just have two minor suggestions:
Keywords: Please try to choose keywords that are not present in the title.
Figure 2C and Figure 3 are hard to read with small letter sizes. Please try to edit these figures to increase their readability.
Author Response
We thank the reviewer for his/her helpful suggestions and comments. We hope we have addressed the points raised by the reviewer. Please find our responses to each comment as reported below:
Reviewer comment 1: Keywords: Please try to choose keywords that are not present in the title.
Answer 1: as suggested by the reviewer, we have removed the key words “wounding” (present in the title) and “Jasmonate” (redundant since there is already “Jasmonic acid”) and added “abiotic stress”, “RNA-Seq” and “transcriptomics” since the focus of the manuscript is mostly on transcriptional profiling during post-harvest. We have decided to keep “apple superficial scald” since these are key elements of the topic of the manuscript. We hope that this choice will meet the reviewer’s approval.
Reviewer comment 2: Figure 2C and Figure 3 are hard to read with small letter sizes. Please try to edit these figures to increase their readability.
Answer 2: we have increased the size of the figures/fonts to make them more readable, as suggested by the reviewer and we have included the new versions of figures in the manuscript. However, in some cases, the figures and fonts are automatically generated by the GO programmes. We have also changed Figure 5 by removing the ID of the genes that were all collapsed on the right hand side of the picture, appearing a black stripes, and were unnecessary. We hope that the new Figures are now satisfactory.